# Effects of the Addition of *Dendrobium officinale* on Beer Yeast Fermentation

Xiaolu Chen [1], Linqiu Li [1], Hailong Yang [1,2,*] and Huabin Zhou [1,2]

1   College of Life & Environmental Science, Wenzhou University, Wenzhou 325035, China
2   Zhejiang Provincial Key Laboratory for Water Environment and Marine Biological Resources Protection, Wenzhou University, Wenzhou 325035, China
*   Correspondence: yanghl99@163.com

**Abstract:** *Dendrobium officinale* is a precious Chinese medicinal plant that is rich in polysaccharides, flavonoids, polyphenols, and other bioactive ingredients, and has a variety of biological activities. To explore the effects of *D. officinale* on the growth and metabolism of *Saccharomyces cerevisiae*, different concentrations (0, 10, 30, 50, and 100 g/L) of fresh *D. officinale* were added to the wort during the fermentation. The amount of yeast, alcohol content, reducing sugars, total acidity, pH, $CO_2$ loss, and foam height were analyzed. Meanwhile, the glucose uptake, cell viability, key enzyme activity of yeast, total phenolics, antioxidant activity, volatile compounds, and consumer acceptance of brewed samples were also analyzed. The results showed that the growth and metabolism of yeast could be promoted by a suitable dosage of *D. officinale* but were inhibited at high dosage (100 g/L). The addition of *D. officinale* increased the activities of glucose-6-phosphate dehydrogenase and alcohol dehydrogenase, while the highest concentration of *D. officinale* (100 g/L) decreased the glucose uptake and cell activity of the yeast. The contents of total phenolics and esters, along with the scavenging activity against ABTS radicals, were increased, indicating that the antioxidant activity and aromatic characteristics of beer would be improved by the addition of *D. officinale*.

**Keywords:** *Dendrobium officinale*; beer; yeast; growth and metabolism; cell viability

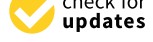



## 1. Introduction

Beer is a low-strength alcoholic beverage consumed in large quantities worldwide, and China ranks first in the world in beer production and consumption [1]. Generally, beer is fermented from cereal materials (mainly barley malt), water, and hops by yeast, and lager beers—which are produced via bottom-fermenting at low temperatures (3.3 to 13.0 °C)—account for more than 90% of the beer produced worldwide [2]. However, more and more drinkers are interested in beers with different flavors, aromas, etc., resulting in sales of craft beer growing faster than those of industrial, lager-style beer in recent decades [3]. Since the 1990s, craft beers have emerged in the US and have developed quickly in the Americas and Europe [3]. In China, the consumption of craft beer has increased by 40% every year since 2015 [4]. Craft beers are not only produced in classic beer styles, but also brewed with new gustatory, olfactory, and visual stimuli through the addition of fruits, spices, and other natural foods [1,5,6]. In particular, the phenolic contents and antioxidant activity of the beer can be increased by the introduction of edible plant materials such as olive leaves, green tea, chestnut, etc. [7–9].

*Dendrobium officinale*, a perennial herb belonging to the *Orchidaceae* family and the *Dendrobium* genus, is a famous and precious medicinal plant in China. Traditionally, it has been used in Chinese medicine to reduce fever, nourish the stomach, promote the secretion of saliva, and prolong the life [10]. Modern pharmaceutical findings show that it is rich in polysaccharides, alkaloids, flavonoids, polyphenols, and other bioactive ingredients, and has a variety of biological activities, e.g., antioxidant, anti-fatigue, immune-enhancing, anticancer, anti-inflammatory, hypoglycemic, hypolipidemic, etc. [11,12]. To

date, *D. officinale* has been artificially planted in Zhejiang, Jiangxi, Guangdong, Yunnan, Guizhou, and other provinces of China, and has been used as a health food approved by the China Food and Drug Administration (CFDA) [13]. Considering the high health value and the successful artificial planting of *D. officinale*, the objective of this work was to explore the use of *D. officinale* in the preparation of beer. For this purpose, fresh *D. officinale* was added to boiling wort, and the physicochemical characteristics and antioxidant activity of the obtained beer were analyzed. Furthermore, the effects of *D. officinale* on the growth and metabolism of yeast were also preliminarily investigated.

## 2. Materials and Methods

### 2.1. Materials and Reagents

Fresh *D. officinale* (moisture content 65.09%) was purchased from LvFeng Tiepishihu Planting Cooperatives in Yueqing, Zhejiang Province, China. Barley malt and *Saccharomyces cerevisiae* CICC 1921 were obtained from Shuanglu Beer Co. (Wenzhou, China). 2,2′-Azino-bis (2-ethylbenzothiazoline-6-sulfonic acid) diammonium salt (ABTS), 2-octanol, fluorescein diacetate (FDA), and 2,3,5-triphenyltetrazolium chloride were purchased from Merck (Shanghai, China) Co., Ltd. Folin–Ciocâlteu reagent, dimethyl sulfoxide, glucose, peptone, and other chemicals were obtained from Sinopharm Chemical Reagent Co., Ltd. (Shanghai, China).

### 2.2. Brewing Process

The flowchart of the brewing process is outlined in Figure 1. The malt was mashed and saccharified. Spent gains were removed by filtration, and the filtered wort was divided into 5 batches of 1000 mL and individually boiled with different concentrations of *D. officinale* (0, 10, 30, 50, and 100 g/L) for one hour. The wort concentration was adjusted to 12 Brix, cooled, and fermented at 11 °C for 7 d after inoculation with $1 \times 10^7$ yeast/mL. Accordingly, the samples were designated as T0 (control), T1, T3, T5, and T10, respectively, and the experiments were repeated in triplicate.

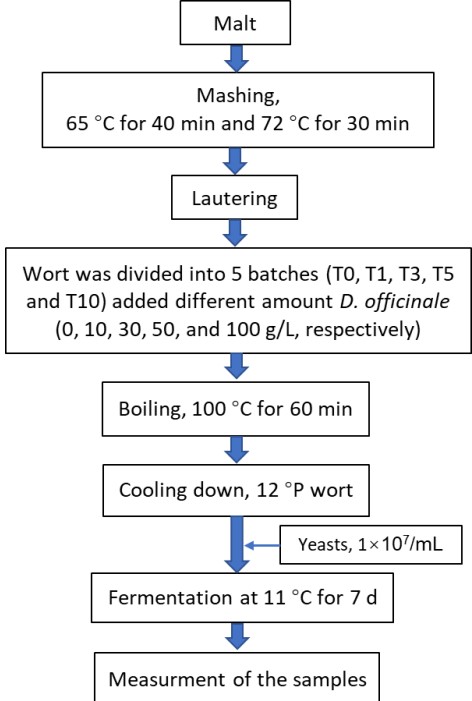

**Figure 1.** Flowchart of the brewing process.

### 2.3. Measurements of pH, Total Acidity, CO$_2$ Loss, Foam Height, Yeast Number and Reducing Sugar and Alcohol Concentrations

The pH was determined using a FiveEasy Plus pH meter (Mettler Toledo (Shanghai) Co., Ltd., Shanghai, China). Total acidity was determined by titration to pH 8.2 with 0.05 mol/L NaOH, and the results were expressed as grams of lactic acid per 100 mL of sample. CO$_2$ loss was determined by measuring the weight of the samples before and after fermentation, and reported as grams per 100 mL of sample. For determination of the yeast number, the sample was diluted and the yeasts were counted using a hemocytometer. The reducing sugar content was estimated by a 3,5-dinitryl-salicylic acid (DNS) colorimetry assay using glucose as the standard. The alcohol concentration was determined using an alcoholmeter (Huangjiapu Glass Instrument Factory, Yuyao, China) after distillation.

For the determination of foam height, the fermentation was performed in tubes (15 × 150 mm) and the foam height was monitored every 12 h. The highest foam height was reported for each sample.

### 2.4. Volatile Compounds

The volatile compounds were analyzed using headspace–solid-phase microextraction–gas chromatography–mass spectrometry (HS–SPME–GC–MS), as described previously [14,15], with some modifications. Briefly, 8 mL of sample and 50 μL of internal standard (2-octanol, 100 mg/L) were added to a 15 mL SPME glass vial with 2.5 g of sodium chloride, which was tightly capped and kept at 80 °C for 15 min to equilibrate. The volatile compounds were extracted and adsorbed using an SPME fiber coated with divinylbenzene/carbonex/polydimethylsiloxane (DVB/CAR/PDMS, 50/30 μm) (Supelco, Bellefonte, PA, USA) for 40 min at the same temperature. Afterwards, the fiber was retracted and immediately inserted into the injection port of an Agilent GC–MS system equipped with a DB-WAX column (30 m × 250 μm × 0.25 μm; Agilent J&W Scientific, Folsom, CA, USA) and desorbed for 5 min at 250 °C. The GC was carried out using helium as a carrier gas at a flow rate of 1 mL/min, and the oven temperature was programmed as follows: 40 °C at the beginning, increased at 5 °C/min to 90 °C, and then to 230 °C at 10 °C/min, and finally maintained at 230 °C for 10 min. For the MS system, the electron energy was 70 eV and the temperatures of the transfer line, quadrupole, and ionization source were 230, 150, and 230 °C, respectively. The mass spectra were obtained using the full scan mode with a scan range of 20–500 *m/z*. The volatile compounds were identified by comparing their mass spectra with those in the NIST11 database.

### 2.5. Consumer Test

For the consumer acceptance test, samples were matured at 4 °C for 1 month. The sensory evaluation was carried out by the scoring method described previously [16,17], with some modifications. In brief, the samples were served in random coded plastic cups, and 30 participants (19 to 35 years old) evaluated the attributes of color, aroma, taste, and overall acceptance of each sample on a nine-point hedonic scale (from 1 = highly disliked to 9 = highly liked).

### 2.6. Determination of Glucose Uptake by Yeast

The glucose uptake by yeast was determined as described by Somnath et al. [18], with some modifications. Firstly, beer yeast was cultured in YPD medium at 28 °C for 7 d, collected by centrifugation (4000 rpm, 5 min), and repeatedly washed using normal saline until no glucose was determined in the supernatant. Afterwards, a 10% yeast suspension was prepared in normal saline. The extracts of *D. officinale* were added to glucose solution (8%, 5 mL), incubated at 30 °C for 10 min, and then 0.5 mL of yeast suspension was added, mixed, and further incubated at 30 °C for 30 min. After that, the supernatant was collected by centrifugation, and the glucose concentration was determined by 3,5-dinitryl-salicylic acid (DNS) colorimetry assay. The glucose uptake by yeast was calculated from the glucose

concentrations before and after incubation, and expressed as milligrams of glucose per gram of fresh yeast biomass per minute (mg/g/min).

### 2.7. Determination of the Cell Viability of Yeast

The yeast viability assay was performed as described by Shi et al. [19], with some modifications. Samples (0.06 g of fresh yeast) were homogenized in the dark with 1.5 mL of 2,3,5-triphenyltetrazolium chloride (1% in 0.1 mol/L phosphate buffer) using an IKA T10 basic homogenizer (IKA, Staufen, Germany) for 5 min. Afterwards, the precipitate was collected by centrifugation, mixed with 2 mL of dimethyl sulfoxide, and centrifuged, and the optical densities of the supernatants were recorded at 485 nm.

To visualize the yeast's viability, samples (0.2 g of fresh yeast) were washed twice with normal saline and stained with FDA (300 μL, 100 μmol/L) for 10 min at room temperature. The samples were then examined using a fluorescence microscope (excitation 488 nm; emission 525 nm).

### 2.8. Determination of Glucose-6-Phosphate Dehydrogenase and Alcohol Dehydrogenase Activities

Yeasts were collected by centrifugation, washed twice with normal saline, and ultrasonically extracted with Tris-HCl buffer (100 mmol/L, pH 7.0, containing 1 mmol/L dithiothreitol, 10 mmol/L $MgCl_2$, and 1 mmol/L EDTA) in an ice bath for 10 min. After centrifugation (12,000 rpm, 4 °C), the supernatant was collected and used for enzyme analysis. Glucose-6-phosphate dehydrogenase analysis (G6PDH) was performed as described by Yuan et al. [20], with some modifications. Briefly, 2.7 mL of reaction solution contained 100 mmol/L Tris-HCl buffer (pH 7.0), 5 mmol/L glucose-6-phosphate, 5 mmol/L $MgCl_2$, and 5 mmol/L NADP. The reaction was started by the addition of 0.3 mL of crude enzyme. One unit of G6PDH activity was defined as the amount of enzyme catalyzing the reduction of 1 μmol of NADP/min, and the result was expressed as units per gram of fresh yeast. Alcohol dehydrogenase (ADH) activity was determined as described by van Rijswijck et al. [21], with some modifications. Briefly, 40 μL of NAD + (50 mmol/L) and 40 μL of ethanol were added to 3 mL of Tris-HCl buffer (100 mmol/L, pH 8.5) and mixed, and the reaction was started by the addition of 0.3 mL of crude enzyme. One unit of ADH activity was defined as the amount of enzyme catalyzing the production of 1 μmol of NADH/min, and the result was expressed as units per gram of fresh yeast.

### 2.9. Determination of Total Phenolics and Antioxidant Activity

The total phenolic content was analyzed colorimetrically by the Folin–Ciocâlteu method using gallic acid as a standard, as described by Nardini and Foddai [8], and the results were expressed as milligrams of gallic acid equivalents per liter of sample.

The antioxidant activity of the samples was determined according to the scavenging ability against ABTS radicals, as described by Nardini and Foddai [8]. The percentage inhibition of absorbance was calculated with reference to an ascorbic acid calibration curve, and the results were expressed as micrograms of ascorbic acid equivalents per liter of sample.

### 2.10. Statistical Analysis

The experimental results are presented as means $\pm$ standard deviations (SD). Duncan's multiple range test was performed to compare the differences using one-way ANOVA in STATISTICA 6.0, and $p < 0.05$ was considered statistically significant.

## 3. Results

### 3.1. Effects of D. officinale on the Growth and Metabolism of S. cerevisiae

First, 0, 10, 30, 50, and 100 g/L of *D. officinale* were added to the wort, and the analysis was performed after 7 days of fermentation. Compared with the control group, the amount of yeast, alcohol content, and $CO_2$ loss were increased in samples T1, T3, and T5; particularly significant ($p < 0.05$) differences in these indices were determined in samples T3 and T5.

Accordingly, the contents of residual reducing sugars were decreased in these samples. However, no significant ($p > 0.05$) differences in the amount of yeast, alcohol content, or $CO_2$ loss were observed between the control (T0) and T10 (Table 1). These results indicate the promoting effect of *D. officinale* at suitable concentrations (10–50 g/L) on the growth and metabolism of beer yeast, but inhibitory effects at the higher concentration (>100 g/L). In addition, the foam height was increased with the increase in the concentration of added *D. officinale*, due to the rich polysaccharides therein.

**Table 1.** Effects of *D. officinale* concentration on the growth and metabolism of *Saccharomyces cerevisiae* *.

| Parameters | Samples | | | | |
|---|---|---|---|---|---|
| | Control (T0) | T1 | T3 | T5 | T10 |
| Reducing sugars (g/L) | 10.47 ± 3.63 [a] | 9.56 ± 0.70 [b] | 8.87 ± 0.21 [b] | 7.15 ± 0.20 [bc] | 9.14 ± 1.25 [c] |
| The number of yeasts (×$10^7$ yeast/mL) | 5.01 ± 0.83 [c] | 7.75 ± 0.88 [ab] | 9.83 ± 0.65 [a] | 9.12 ± 1.33 [a] | 5.49 ± 0.10 [bc] |
| Total acidity (g/100 mL) | 2.71 ± 0.07 [a] | 1.74 ± 0.12 [c] | 1.60 ± 0.04 [cd] | 1.50 ± 0.07 [d] | 2.01 ± 0.11 [b] |
| pH | 4.54 ± 0.02 [b] | 4.40 ± 0.01 [d] | 4.41 ± 0.02 [d] | 4.46 ± 0.02 [c] | 4.67 ± 0.02 [a] |
| Alcohol (%) | 4.70 ± 0.12 [bc] | 5.87 ± 0.31 [ab] | 6.27 ± 0.12 [a] | 6.30 ± 0.10 [a] | 4.10 ± 1.25 [c] |
| $CO_2$ loss (g/100 mL) | 3.49 ± 0.25 [cd] | 4.22 ± 0.04 [bc] | 4.65 ± 0.05 [ab] | 5.41 ± 0.75 [a] | 2.95 ± 0.10 [d] |
| Foam height (mm) | 1.50 ± 0.50 [d] | 3.67 ± 0.42 [c] | 5.43 ± 0.49 [b] | 5.83 ± 0.35 [b] | 9.10 ± 0.10 [a] |

* T0, T1, T3, T5, and T10 were the samples with 0, 10, 30, 50, and 100 g/L of *D. officinale*, respectively. Different letters in a row indicate significant differences between different groups at $p < 0.05$.

### 3.2. Effects of D. officinale on the Glucose Uptake and Cell Viability of S. cerevisiae

*D. officinale* was ultrasonically extracted with deionized water, and the extracts equivalent to 30 and 100 g/L of fresh materials were used for the experiment. As shown in Figure 2, the glucose uptake ability of *S. cerevisiae* was decreased by the addition of *D. officinale*, and the decrease was insignificant ($p > 0.05$) at 30 g/L but significant ($p < 0.05$) at 100 g/L, indicating that high concentrations of *D. officinale* would inhibit the glucose uptake of *S. cerevisiae*.

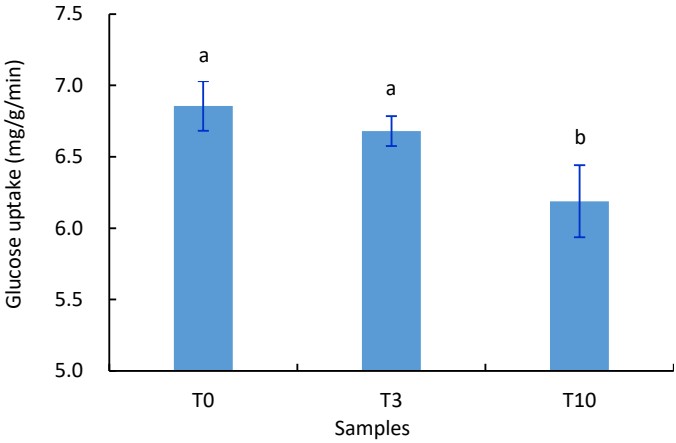

**Figure 2.** Effects of *D. officinale* on the glucose uptake of *S. cerevisiae*. T0, T3, and T10 were the samples with 0, 30, and 100 g/L of *D. officinale*, respectively. Different letters indicate significant differences at $p < 0.05$.

The effects of *D. officinale* on the cell viability of *S. cerevisiae* are presented in Figure 3. Similar to the results of glucose uptake, there was no significant difference between the viability values of T0 and T3, while the viability value of T10 was significantly ($p < 0.05$) decreased by 48.70% as compared with that of the control (T0) (Figure 3A).

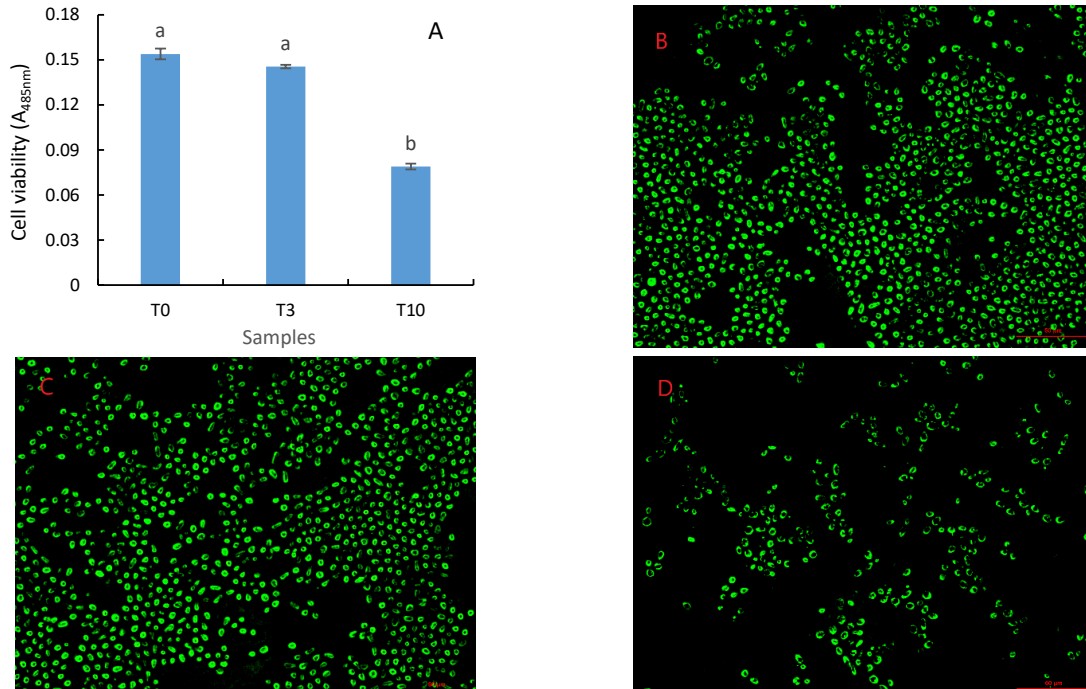

**Figure 3.** Effects of *D. officinale* on the cell viability of *S. cerevisiae*. T0, T3, and T10 were the samples with 0, 30, and 100 g/L of *D. officinale*, respectively. Different letters indicate significant differences at $p < 0.05$. (**A**): value of cell viability; (**B**): T0; (**C**): T3; (**D**): T10.

To visually demonstrate the effects of *D. officinale* on the cell viability of *S. cerevisiae*, cells were stained with FDA and observed using a fluorescence microscope. As shown in Figure 3B–D, the fluorescence intensity (green pots) was strong in the T0 and T3 samples, while only slight fluorescence was observed in T10. This result is consistent with the viability values, revealing the inhibitory effect of *D. officinale* on *S. cerevisiae* at high concentrations.

### 3.3. Effects of D. officinale on the ADH and G6PDH Activities of S. cerevisiae

As shown in Figure 4, the ADH and G6PDH activities of *S. cerevisiae* were increased by the addition of *D. officinale*. Compared with the control (T0), the ADH activity was increased 1.84- and 2.25-fold, and the G6PDH activity was increased 1.74- and 1.76-fold, in samples T3 and T10, respectively.

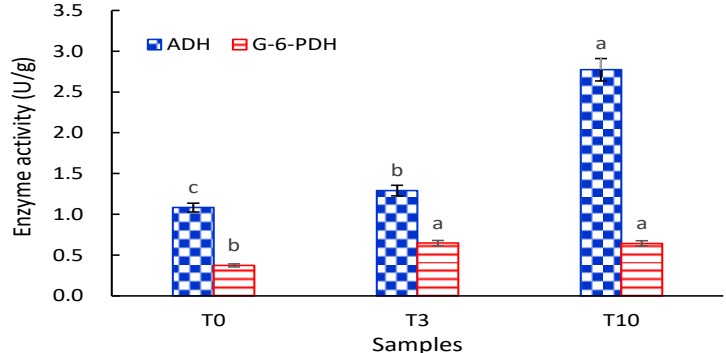

**Figure 4.** Effects of *D. officinale* on the activities of alcohol dehydrogenase (ADH) and glucose-6-phosphate dehydrogenase (G6PDH) in *S. cerevisiae*. Different letters indicate significant differences between different samples at $p < 0.05$.

### 3.4. Total Phenolic Content and Antioxidant Activity

As shown in Figure 5, the total phenolic content of beer samples was increased by the addition of *D. officinale*. Among them, no significant ($p > 0.05$) difference in the total phenolic content was determined between T3 (0.62 mg/mL) and T0 (control, 0.60 mg/mL), while the total phenolic content in T10 (0.68 mg/mL) was significantly ($p < 0.05$) higher than that of the control—by 12.65%. The antioxidant activity evaluated by the ABTS radical cation decolorization assay showed higher values in the samples with added *D. officinale* (T3 and T10) as compared with the control (T0, 1.91 µg AsA/mL). As with the total phenolic content, the increase in ABTS radical scavenging ability was significant ($p < 0.05$) in T10 (2.62 µg AsA/mL), while it was insignificant ($p > 0.05$) in T3 (2.04 µg AsA/mL).

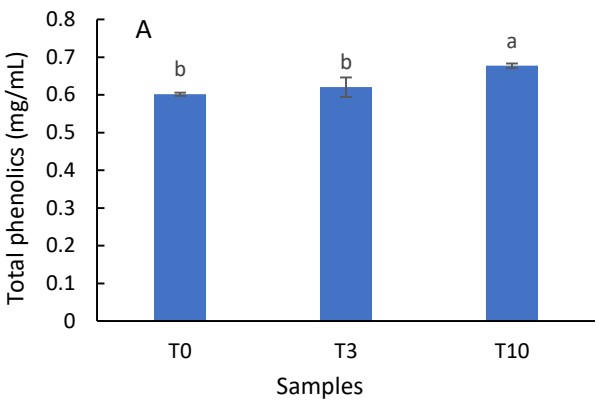 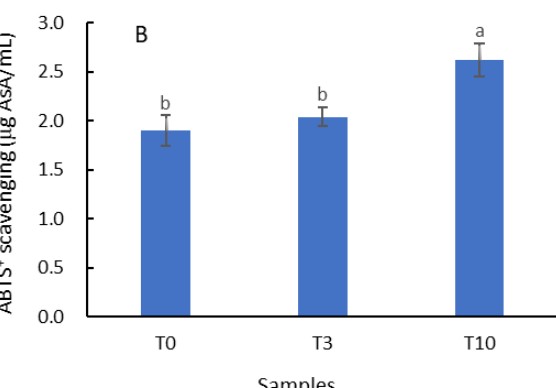

**Figure 5.** Effects of the addition of *D. officinale* on the total phenolic content (**A**) and antioxidant activity (**B**) of beer samples. Different letters indicate significant differences between different samples at $p < 0.05$.

### 3.5. Consumer Acceptance

Figure 6 demonstrates the sensory evaluation of beer samples fermented with different concentrations of *D. officinale*. T3 showed the highest sensory score (75.33), while the use of higher concentrations of *D. officinale* led to higher polyphenol extraction, resulting in lower acceptance. Therefore, the addition of 30 g/L *D. officinale* is suitable for craft beer production.

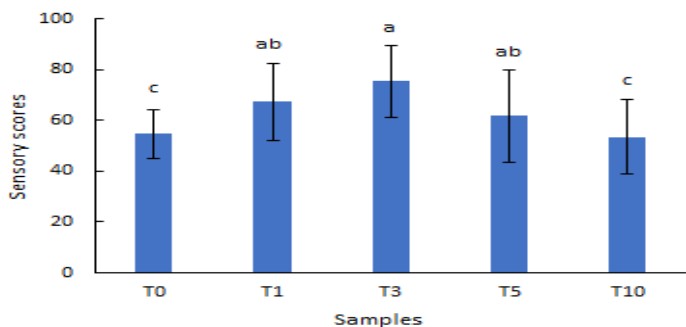

**Figure 6.** Sensory evaluation of beer samples fermented with *D. officinale*. T0, T1, T3, T5, and T10 were the samples with 0, 10, 30, 50, and 100 g/L of *D. officinale*, respectively. Different letters indicate significant differences at $p < 0.05$.

### 3.6. Volatile Compounds

As shown in Table 2, in addition to ethanol, 28 volatile compounds were identified in T0, including alcohols (6 compounds), esters (7 compounds), acids (8 compounds), aldehydes (3 compounds), and others (4 compounds). The addition of *D. officinale* increased the components of volatile compounds, and 38 volatile compounds were identified in T3, including alcohols (8 compounds), esters (11 compounds), acids (9 compounds), aldehydes

(4 compounds), and others (6 compounds). Compared with the control (T0), the contents of esters were increased, while those of alcohols, acids, and aldehydes were decreased. Among them, the concentrations of 3-methyl-1-butanol acetate and octanoic acid ethyl ester in T3 were 5.43- and 3.70-fold higher, respectively, while the concentrations of 3-methyl-1-butanol, phenylethyl alcohol, hexanoic acid, and octanoic acid in T3 were 3.71-, 4.26-, 4.09-, and 2.70-fold lower than those in the control, respectively.

**Table 2.** Effects of the addition of *D. officinale* on the contents of individual volatile components of beer samples (mg/L) *.

| Number | Compound | Molecular Weight | CAS Number | Samples | |
|---|---|---|---|---|---|
| | | | | T0 | T3 |
| | Alcohols | | | | |
| 1 | 1-Propanol | 60.06 | 000071-23-8 | 0.051 | 0.010 |
| 2 | 1-Propanol, 2-methyl- | 74.07 | 000078-83-1 | 0.147 | 0.042 |
| 3 | 1-Butanol, 3-methyl- | 88.09 | 000123-51-3 | 4.403 | 1.188 |
| 4 | 1,6-Octadien-3-ol, 3,7-dimethyl- | 154.14 | 000078-70-6 | - | 0.006 |
| 5 | 1-Octanol | 130.14 | 000111-87-5 | - | 0.010 |
| 6 | 2-Furanmethanol | 98.04 | 000098-00-0 | 0.144 | 0.028 |
| 7 | 3-Methoxybenzyl alcohol | 138.07 | 006971-51-3 | - | 0.011 |
| 8 | Benzyl alcohol | 108.06 | 000100-51-6 | 0.005 | - |
| 9 | Phenylethyl alcohol | 122.07 | 000060-12-8 | 1.829 | 0.429 |
| | Esters | | | | |
| 10 | 1-Butanol, 3-methyl-, acetate | 130.1 | 000123-92-2 | 0.042 | 0.228 |
| 11 | Hexanoic acid, ethyl ester | 144.12 | 000123-66-0 | 0.016 | 0.036 |
| 12 | Octanoic acid, ethyl ester | 130.14 | 006169-06-8 | 0.090 | 0.333 |
| 13 | Undecanoic acid, ethyl ester | 172.15 | 000106-32-1 | - | 0.005 |
| 14 | 2-Furanmethanol, acetate | 140.05 | 000623-17-6 | 0.024 | 0.008 |
| 15 | Decanoic acid, ethyl ester | 200.18 | 000110-38-3 | - | 0.003 |
| 16 | Acetic acid, 2-phenylethyl ester | 164.08 | 000103-45-7 | - | 0.155 |
| 17 | Acetic acid, phenyl ester | 136.05 | 000122-79-2 | 0.007 | - |
| 18 | Tributyl phosphate | 266.17 | 000126-73-8 | - | 0.004 |
| 19 | 9,12-Octadecadienoic acid, methyl ester | 294.26 | 002566-97-4 | 0.009 | - |
| 20 | 1,2-Benzenedicarboxylic acid, bis(2-methylpropyl) ester | 278.15 | 000084-69-5 | - | 0.007 |
| 21 | Dibutyl phthalate | 278.15 | 000084-74-2 | 0.037 | 0.015 |
| 22 | Hexanedioic acid, bis(2-ethylhexyl) ester | 370.31 | 000103-23-1 | - | 0.007 |
| | Acids | | | | |
| 23 | Acetic acid | 60.02 | 000064-19-7 | - | 0.029 |
| 24 | Butanoic acid | 88.05 | 000107-92-6 | 0.012 | - |
| 25 | Hexanoic acid | 116.08 | 000142-62-1 | 0.319 | 0.078 |
| 26 | Octanoic acid | 144.12 | 000124-07-2 | 1.979 | 0.733 |
| 27 | Nonanoic acid | 158.13 | 000112-05-0 | 0.041 | 0.017 |
| 28 | n-Decanoic acid | 172.15 | 000334-48-5 | 0.258 | 0.020 |
| 29 | Benzoic acid | 122.04 | 000065-85-0 | 0.027 | 0.007 |
| 30 | 9-Decenoic acid | 170.13 | 014436-32-9 | 0.084 | 0.019 |
| 31 | Dodecanoic acid | 200.18 | 000143-07-7 | 0.037 | 0.003 |
| 32 | n-Hexadecanoic acid | 256.24 | 000057-10-3 | - | 0.015 |
| | Aldehydes | | | | |
| 33 | Furfural | 96.02 | 000098-01-1 | 0.024 | - |
| 34 | Benzaldehyde | 106.04 | 000100-52-7 | 0.035 | 0.002 |
| 35 | Benzeneacetaldehyde | 120.06 | 000122-78-1 | 0.051 | 0.014 |
| 36 | Benzaldehyde, 2,4-dimethyl- | 134.07 | 015764-16-6 | - | 0.014 |
| 37 | Benzaldehyde, 2,4,6-trimethyl- | 148.09 | 000487-68-3 | - | 0.016 |
| | Others | | | | |
| 38 | Isomaltol | 126.03 | 003420-59-5 | 0.069 | 0.010 |
| 39 | 1-Propanol, 3-(methylthio)- | 106.05 | 000505-10-2 | 0.005 | - |
| 40 | Ethanone, 1-(1H-pyrrol-2-yl)- | 109.05 | 001072-83-9 | - | 0.003 |
| 41 | 2(3H)-Furanone, dihydro-5-pentyl- | 156.12 | 000104-61-0 | - | 0.003 |
| 42 | 2-Methoxy-4-vinylphenol | 150.07 | 007786-61-0 | 0.178 | 0.056 |
| 43 | Phenol, 2,4-bis(1,1-dimethylethyl)- | 206.17 | 000096-76-4 | - | 0.06 |
| 44 | Benzofuran, 2,3-dihydro- | 120.06 | 000496-16-2 | - | 0.009 |
| 45 | Indole | 117.06 | 000120-72-9 | 0.016 | - |

* T0 and T3 were the samples with 0 and 30 g/L of *D. officinale*, respectively.

## 4. Discussion

Craft beer is often brewed with the addition of suitable adjuncts with unique flavors and/or functional components, and the adjuncts may affect the growth and metabolism of beer yeast when added in the wort-boiling or fermentation stages [9]. As reported previously, the addition of goji berries at 50 g/L [5] and okra at 1 g/L [22] promoted the ethanol production of beer yeast; there were no effects on ethanol content caused by the addition of mango at 20% [14] and *Parastrephia lucida* leaves at 0.1–5% [23], while the addition of hawthorn juice or fruits at 10% resulted in the opposite effect [16]. In this work, *D. officinale* was added in the wort-boiling stage and showed promoting effects on yeast growth and ethanol production at suitable doses; however, a decreased promoting effect—or even an inhibitory effect—was demonstrated at higher doses (Table 1), due to the existence of secondary metabolites such as phenolics, anthraquinones, and alkaloids [12]. The main biochemical process of beer brewing is that yeast absorbs fermentable sugars, converting them into ethanol through glycolysis, and produces other metabolites concomitantly. Glucose uptake and cell viability experiments also showed non-significant effects of the addition of *D. officinale* at low dosages but inhibitory effects at high dosages (Figures 2 and 3).

Beer contains a variety of phenolic compounds originating from barley and hops, and it shows antioxidant activity [6,24]. The addition of plant adjuncts for craft beer brewing may increase the phenolic contents and antioxidant activity [9]. Thyme, juniper, and lemon balm were added to beer samples, and the total phenolic contents were increased by 37.09%, 30.36%, and 29.55%, respectively [25]. Gasiński et al. [16] reported that the contents of polyphenolic compounds and the ABTS and DPPH radical scavenging abilities were increased by more than 2.0-fold, 2.0-fold, and about 6.0-fold, respectively, in beers brewed with the addition of dotted hawthorn (*Crataegus punctata*) juice. Results from another work by Gasiński et al. [14] showed that the total polyphenol contents and the ABTS and DPPH radical scavenging activities in beer were increased by 42.8%, 44.3%, and 42.4%, respectively, with the addition of mango juice. Recently, Lazzari et al. [17] reported that high levels of total phenolic compounds and antioxidant activity were achieved by the replacement of hops with rubim (*Leonurus sibiricus* L.) and mastruz (*Chenopodium ambrosioides* L.). As a medicinal plant, *D. officinale* contains a variety of phenolic compounds and shows antioxidant activity [12]. Similarly, the increased total phenolic contents and antioxidant activity were also observed in the beer brewed with the addition of *D. officinale* in this work (Figure 5).

G6PDH is the first enzyme of the pentose phosphate pathway, which is important for the biosynthesis of erythrose 4-phosphate and ribose 5-phosphate; therefore, it is related to the proliferation of yeast [26]. The production of G6PDH by *S. cerevisiae* [27] could also be improved by the addition of *D. officinale* extract at suitable dosages. ADH catalyzes the conversion of pyruvate to ethanol; therefore, its activity directly affects the conversion rate of ethanol [26]. Choi et al. [28] reported that *Schizandrae fructus* extract could increase the ADH activity of *S. cerevisiae*. A similar result was also observed with the addition of *D. officinale* extract.

Beer contains a variety of volatile chemicals, the main ones of which are ethanol and carbon dioxide. However, other volatile compounds—including higher alcohols, esters, ketones, aldehydes, organic acids, phenols, and sulfur compounds—play a vital role in the taste and aroma despite being present at very low concentrations [16,29]. The individual volatile chemicals and their contents in beers are affected by the yeast strains, fermentation substrates, brewing conditions, and other factors [30]. Yin et al. [31] compared the effects of amino acids on the production of volatile compounds by beer yeast and found that glutamic acid, aromatic amino acids, and branched-chain amino acids can promote the production of higher alcohols, while leucine, valine, phenylalanine, serine, and lysine can increase the production of esters. Cioch-Skoneczny et al. [32] reported that adding brewing triticale to wort can promote the production of acids by beer yeast. The additives for craft beer can also change the volatile components of beer samples. As reported by Zapata et al. [33], the concentrations of methyl benzoate and ethyl hexanoate were increased, while

the concentrations of 4-ethyl guaiacol, ethyl dodecanoate, and isoamyl octanoate were decreased in beer samples brewed with the addition of quince (*Cydonia oblonga*) fruits. Gasiński et al. [16] also found that the formation of hexyl acetate, 2-methyl butyl ester, 5-methyl ethyl caproate, methyl caproate, and isoamyl caproate was increased in craft beer brewed with the addition of 10% dotted hawthorn (*Crataegus punctata*), resulting in the increase in volatile component contents from 0.359 mg/L to 2.879 mg/L. In this work, the contents of 3-methyl-1-butanol acetate, hexanoic acid ethyl ester, octanoic acid ethyl ester, and acetic acid 2-phenylethyl ester were remarkably increased, leading to an increase in esters, which was similar to the results of [22], who reported that the ester contents in beers were increased by the addition of okra (*Abelmoschus esculentus*). In addition, *D. officinale* contains a series of volatile compounds, among which 22 alcohols and 23 esters were identified [34]. Some components—such as tributyl phosphate, 2,4-di (1,1-dimethylethyl) phenol, and other volatile chemicals—could enter the beer sample during brewing, enriching the flavor of craft beers brewed with *D. officinale*.

## 5. Conclusions

The addition of *D. officinale* could affect the growth and metabolism of beer yeast, and the total phenolic contents and antioxidant activity were increased with the increase in the dosage. Compared with the control, the sample brewed with 30 g/L of *D. officinale* had a higher content of esters, while also receiving the highest scores in the consumer test. In summary, craft beer with good flavor and health functions could be brewed with the addition of *D. officinale* at a suitable dosage.

**Author Contributions:** Conceptualization, H.Y.; investigation, X.C. and L.L.; data curation, X.C. and L.L.; methodology, X.C. and H.Y.; resources, H.Z. and H.Y.; supervision, H.Z. and H.Y.; validation, H.Z. and H.Y.; writing—original draft, X.C. and L.L.; writing—review and editing, H.Y. All authors have read and agreed to the published version of the manuscript.

**Funding:** This research was funded by the Key Research and Development Project of Zhejiang Province (2020C02038) and the New Miao Talent Project of Zhejiang (2021R429032), China.

**Institutional Review Board Statement:** Not applicable.

**Informed Consent Statement:** Informed consent was obtained from all subjects involved in the study.

**Data Availability Statement:** Data is contained within the article.

**Conflicts of Interest:** The authors declare no conflict of interest.

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
