# Peer review of "Effects of the Addition of Dendrobium officinale on Beer Yeast Fermentation"

_fermentation, doi:10.3390/fermentation8110595_

Round 1

Reviewer 1 Report

Dear Authors,

I would like to underline some new ideas presented in the article and related to beer additives. I do acknowledge that consumers look into new tastes and additional benefits from beverages. Still, I noticed some weak points which should be improved:

1. Please describe in the abstract what D. officinale is and add one or two sentences descrbing the purpose of adding this herb into the wort.

2. In the introduction section there are data related to China. The journal is adressed to worldwide readers. Please add information about other regions and trends in craft beer making.

3. Describe what yeast did you tested (line 50).

4. Please add information which S. cerevisiae strain was used.

5. I wonder if boilng the wort with the herb did not affect biological functions of some compunds present in the plant extract.

6. Please add yeast "biomass" line 122

7. There should be provided a definition of 1 U related to 1 mikromole of substrate transformed or product synthesized.

8. What post-hoc tests were used for statistical analysis? Did normality tests were made? Please  provide details in materials and methods section.

9. I have some doubts wheter a normal distribution was achieved in relation to pH in Table 1.

10. Please write Dendrobium italics (line 228, 249).

11. Past tense should be provided in the conclusions sections.

12. Please comment why D. officinale could promote yeast growth at certain dose? What is your hypothesis?

13. There is no explanation in the discussion section why enzyme activities were affected by D. officinale?

14. How was total phenolic acid in the herb extract before boiling? How high tmeperature affected antioxidant activity?

Reviewer 2 Report

The manuscript is well written, the rationale is clear. Results are sound and clearly exposed.

Few minor changes should be done as follows:

- paragraph 3.4, lines 214-221: the numeric values for TPC and ABTS scavenging activity should be done in the text to allow comparison with data from literature. 

- If it is possible, at least for fig. 5 the values referring to T5 could be shown.

- paragraph 3.6, line 239: "T0" is wrong. It should be replaced with "T3".

- Discussion, line 255: there is a typing mistake, the word "mongo" should be replaced by "mango".

Round 2

Reviewer 1 Report

Dear Authors, Thank you very  much for providing  necesarry changes.